# Characteristics of Biofilms Formed by *C. parapsilosis* Causing an Outbreak in a Neonatal Intensive Care Unit

**DOI:** 10.3390/jof8070700

**Published:** 2022-07-01

**Authors:** Atsushi Miyake, Kenji Gotoh, Jun Iwahashi, Akinobu Togo, Rie Horita, Miho Miura, Masahiro Kinoshita, Keisuke Ohta, Yushiro Yamashita, Hiroshi Watanabe

**Affiliations:** 1Department of Infection Control and Prevention, Kurume University School of Medicine, Kurume 831-0011, Japan; miyake_atsushi@kurume-u.ac.jp (A.M.); iwahashi@kurume-u.ac.jp (J.I.); miura-miho@kurume-u.ac.jp (M.M.); hwata@kurume-u.ac.jp (H.W.); 2Department of Pediatrics and Child Health, Kurume University School of Medicine, Kurume 831-0011, Japan; kinoshita_masahiro@kurume-u.ac.jp (M.K.); yushiro@kurume-u.ac.jp (Y.Y.); 3Advanced Imaging Research Center, Kurume University School of Medicine, Kurume 830-0011, Japan; togou_akinobu@kurume-u.ac.jp (A.T.); kohta@kurume-u.ac.jp (K.O.); 4Department of Clinical Laboratory Medicine, Kurume University Hospital, Kurume 831-0011, Japan; horita_rie@kurume-u.ac.jp

**Keywords:** *Candida parapsilosis*, outbreak, neonatal intensive care unit (NICU), infection control, microsatellite analysis, scanning electron microscopy (SEM), biofilm

## Abstract

Background: We dealt with the occurrence of an outbreak of *Candida parapsilosis* in a neonatal intensive care unit (NICU) in September 2020. There have been several reports of *C. parapsilosis* outbreaks in NICUs. In this study we describe our investigation into both the transmission route and the biofilm of *C. parapsilosis*. Methods: *C. parapsilosis* strains were detected in three inpatients and in two environmental cultures in our NICU. One environmental culture was isolated from the incubator used by a fungemia patient, and another was isolated from the humidifier of an incubator that had been used by a nonfungemia patient. To prove their identities, we tested them by micro satellite analysis. We used two methods, dry weight measurements and observation by electron microscopy, to confirm biofilm. Results: Microsatellite analysis showed the five *C. parapsilosis* cultures were of the same strain. Dry weight measurements and electron microscopy showed *C. parapsilosis* formed biofilms that amounted to clumps of fungal cells. Conclusions: We concluded that the outbreak happened due to horizontal transfer through the humidifier of the incubator and that the *C. parapsilosis* had produced biofilm, which promoted an invasive and infectious outbreak. Additionally, biofilm is closely associated with pathogenicity.

## 1. Introduction

In recent years, the rates of mortality for neonates have declined due to remarkable progress in neonatal care in areas such as antenatal steroid administration, postnatal surfactant replacement, and technological interventions. A negative result in neonatal mortality, however, was caused by increases in the rates of infections, which have become the main cause of post-natal death [1,2]. This development underscores the importance of early detection and appropriate therapeutic interventions. Late-onset infections in a neonatal intensive care unit (NICU) are fundamentally hospital-acquired infections, and an environmental infection control strategy is extremely essential. *Candida* species infections are the major causes of outbreaks in NICUs. *Candida albicans* has been the species most often associated with neonatal infection, and *Candida parapsilosis* is the second most-common yeast species isolated from bloodstream infections. The reason for this is that *C. parapsilosis* forms biofilms, which adhere to medical devices such as indwelling catheters, increasing the risk of bloodstream infection. Once the infection develops, it is often difficult to treat because the fungal cells within the biofilm are covered by an extracellular matrix that physically reduces the concentration of antifungal drug in the biofilm and controls growth through quorum sensing mechanisms [3,4,5,6]. On the other hand, since biofilm formation is one of the systems by which fungi survive in the environment, it is known to be a cause of nosocomial infections if proper environmental management is not performed [7].

In September 2020, we detected three nosocomial cases of invasive *C. parapsilosis* infections in the NICU at Kurume University Hospital. Initially, one infant tested positive for *C. parapsilosis* fungemia. After three weeks, the *C. parapsilosis* strain was isolated from the aseptic samples of another two infants, and the Infection Control Team (ICT) intervened. An immediate environmental survey was performed by the ICT. The *C. parapsilosis* strain was isolated from two locations: the door of an incubator used by the fungemia infant and the incubator’s humidifier that was used by the nonfungemia infant. Herein, we clarify the transmission routes, biofilm production, characteristics of biofilm formation, and discuss the steps taken to prevent further infections.

## 2. Materials and Methods

### 2.1. Ethical Approval

All studies described herein were approved by the Human Ethics Review Boards of the Kurume University School of Medicine on 26 April 2021 (number 21021).

### 2.2. Setting and Outbreak Description

In the Kurume University Hospital, there are 25 diagnosis and treatment departments that serve 24 wards with 1018 beds, which includes a Perinatal Medical Center. The Perinatal Medical Center has an NICU with 12 beds, a growing care unit (GCU) with 18 beds, and a milk preparation room. The NICU accepts many severe patients, such as extremely low-birthweight infants (ELBWI), those with congenital heart disease, and patients with pediatric surgical disease. The first case of *C. parapsilosis* was found using a patient’s blood culture on 9 September 2020. About three weeks after the first case, *C. parapsilosis* strains were isolated from the aseptic samples of two patients on 26 and 27 September 2020. The ICT classified the transmission of these infections as an outbreak.

### 2.3. Patients and Environmental Survey

*C. parapsilosis* strains were isolated from three patients of the NICU. We collected data from the medical records of the patients, which included gender, gestational age, birth weight, age of onset, basal diseases, site of isolation, the use of medical devices, antifungal therapies, β-D glucan, and outcomes.

We took 21 samples from five hands of health care workers (HCW) and eight locations on 6 October 2020: HCWs included 3 nurses and 2 doctors who were caring fungemia patients, five samples from the milk-preparation room, 7 samples from 3 infant incubators (Atom Infant incubator Dual Incu I, Atom Medical Corporation, Tokyo, Japan) in the NICU, 1 sample from a milk warmer in the NICU, 1 sample from an infant warmer in the GCU, 1 sample from a milk warmer in the GCU, and 1 sample from a used baby bottle in the GCU. To gather samples, we used microbiological transport swabs. Two *C. parapsilosis* strains were isolated from environmental survey, and in total 5 *C. parapsilosis* strains were included in this study.

### 2.4. Identification Test and Antifungal Susceptibility Profile

The samples were inoculated onto Sabouraud agar plates (Eiken, Tokyo, Japan) and incubated at 35 °C. Cultures were considered negative if no growth occurred after 7 days of incubation. Acquired *Candida* were screened by color, texture, and macromorphological aspects. Colony types were identified using a MALDI-TOF MS Biotyper (Bruker Daltonics, Bremen, Germany). We determined the minimum inhibitory concentration (MIC) of the *C. parapsilosis* strains from the 3 patients. The MICs of Amphotercin B (AMPH-B, Sumitomo Pharma, Osaka, Japan), Fluconazole (FLCZ, Pfizer, New York City, NY, USA), Micafungin (MCFG, Astellas Pharma, Tokyo, Japan), Voriconazole (VRCZ, Pfizer, New York City, NY, USA), and Caspofungin (CPFG, MSD, Rahway, NJ, USA) were determined using a commercial Frozen Plate for the Antifungal Susceptibility Testing of Yeasts (Eiken, Tokyo, Japan). In the drug susceptibility testing of the target strains, those with the MIC to FLCZ, MCFG, and CPFG ≥ 8 μg/mL, MIC to VRCZ ≥ 1 μg/mL, and MIC to AMPH-B > 1 μg/mL were judged to be resistant according to the guidelines of the Clinical and Laboratory Standards Institute [8].

### 2.5. Genetic Variability Analysis: Microsatellite Analysis

We performed microsatellite analysis as a method to evaluate the homology of *C. parapsilosis* strains isolated from patients and the environment. Several reports have shown that microsatellite analysis is the highest-resolution method that can be used to study a *C. parapsilosis* outbreak [9,10]. Five *C. parapsilosis* strains were grown at 37 °C for 24 h on potato dextrose agar plates (Becton, Dickinson and Company, Sparks, MD, USA). Genomic DNA was extracted using microLYSIS-PLUS (Gel Company, Inc., San Francisco, CA, USA). PCR was carried out according to the protocol described previously with minor modifications. Four *C. parapsilosis*-specific domains were used, designated as CP1 (forward: 5′-AAA GTG CTA CAC ACG CAT CG-3′, reverse: 5′-GGC TTG CAA TTT CAT TTC CT-3′), CP4 (forward: 5′-CAA ATC ATC CAG CTT CAA ACC-3′, reverse: 5′-CAT CAA ACA AGA ATT CGA TAT CA-3′), CP6 (forward: 5′-CAG GAA CAG GAC AAT GGT GA-3′, reverse: 5′-TCT GGA GCC TCT AGG ACG TTT-3′), and B5 (forward: 5′-AGG TTT GTA GTA GTG TCC CTA TGG-3′, reverse: 5′-TAT CTC TCT CGC CAT TTG AAC G-3′) [11]. Forward primers for CP1 and CP6 were labeled with 6-FAM. Forward primers for CP4 and B5 were labeled with HEX. Microsatellite fragments were synthesized using PrimeSTAR HS DNA polymerase (Takara Bio Inc., Shiga, Japan). PCR reaction mixtures were preheated at 98 °C for 2 min, and amplification was performed by 35 cycles of denaturation at 98 °C for 10 s, annealing at 54–56 °C (depending on the locus) for 15 s, and elongation at 68 °C for 25 s, along with a final extension step at 68 °C for 10 min. The PCR products were analyzed at Macrogen Inc. (Seoul, Korea) by capillary electrophoresis on an ABI3730XL DNA Analyzer (Applied Biosystems, Foster City, CA, USA).

### 2.6. Biofilm Formation and Quantitation

Five *C. parapsilosis* strains were used to form the biofilm. To 1.5 mL potato dextrose medium (PDM, Becton, Dickison and Company, Sparks, MD, USA), 1 × 10^6^ CFU of fungal cells was added using 24-well plates that were then incubated at 37 °C for 40 h. After rinsing with phosphate-buffered saline (PBS; pH 7.5), 1 mL of PBS was added; and cell scrapers (9000-220; AGC Techno Glass, Shizuoka, Japan) were used to collect the biofilms, which were then transferred to microtubes. These samples were centrifuged at 4 °C and 15,000 rpm for 5 min. The supernatants were removed, and the precipitates were treated with ethanol at 4 °C for 5 min then centrifuged at 15,000 rpm for 5 min. The supernatants were removed, and the precipitates were dried in a heat block at 70 °C for 5 h in a safety cabinet with a fan running. The dried materials were weighed using a semi-micro balance. This experiment was carried out three times, and three biofilms for each strain were formed in each assay.

### 2.7. Electron Microscopy

The biofilm formation of two *C. parapsilosis* strains was assessed via scanning electron microscopy (SEM). The biofilm formation was initiated on a glass piece with 1 × 10^6^ CFU of fungal cells in 1.5 mL of PDM for each well of a chamber slide (CultureSlides, FALCON, Big Flats, NY, USA), and the slide was incubated at 37 °C for 48 h. After removing the PDM, the biofilms were rinsed with PBS (pH 7.5). The specimens on the chamber slide were fixed with 2% glutaraldehyde in PBS for 1 h at room temperature, rinsed with PBS, and then further fixed with 1.5% potassium ferrocyanide and 2% osmium tetroxide for 1 h at 4 °C. After rinsing with distilled water, the specimens were reacted with 1% thiocarbohydrazide, rinsed with distilled water, immersed in a 2% osmium tetroxide solution for 1 h at 4 °C, and washed again with distilled water. The specimens were continuously dehydrated with ethanol graded from 50% to 90% and transferred to 100% t-butyl alcohol for 3 changes at room temperature. The specimens in t-butyl alcohol were placed in a refrigerator, the t-butyl alcohol was frozen, and the specimens were then freeze-dried (EIKO ID-2, EIKO, Aichi, Japan). The dried specimens on a chamber slide were coated with osmium metal for surface conduction (NeoC Pro, Meiwa Fosis, Tokyo, Japan) prior to SEM examination (JSM IT800, JEOL).

## 3. Results

### 3.1. Patients Characteristics and Environmental Survey

We diagnosed patient 1 (P1) and patient 3 (P3) with bloodstream infections and patient 2 (P2) as having a urinary-tract infection. As P2 developed a high fever, had a high level of β-D glucan, and had urinary tract malformation, we diagnosed her with an invasive fungal infection. The clinical characteristics of the three patients are summarized in Table 1. P1 and P3 had peripherally inserted central catheters (PICCs) and gastrointestinal drain because of peritonitis. P2 was a case of VACTERL association (the occurrence of at least three conditions: vertebral defects, anal atresia, cardiac defects, tracheo-esophageal fistula, renal anomalies, and limb anomalies). P2 had a urethral catheter.

An environmental survey isolated *C. parapsilosis* from two places. One place was P3’s incubator (E1) and the second place was the humidifier of the nonfungemia infant’s incubator (E2). *C. parapsilosis* was not isolated from the hands of the HCWs. Figure 1 is a picture of E2. Figure 2 shows our NICU ward layout and the locations where the samples of *C. parapsilosis* were detected.

### 3.2. MIC

The results of the MIC showed that all the *C. parapsilosis* isolated from three patients were susceptible to CPFG, MCFG, VRCZ, and FLCZ (Table 2). 

### 3.3. Genetic Variability Analysis: Microsatellite Analysis

All specimens showed the same genotype within microsatellite loci of CP1, CP6, and B5. However, a specimen from P2 with different collection dates showed only a 324 bp fragment in locus CP4, whereas other specimens were heterozygous (305 and 324 bp fragments). It is probable that mutations occurred in the primer-annealing region of the 305 bp fragment, which would have resulted in a failure to amplify the fragment. This result suggests that a single strain could have initially spread to pediatric wards, and then a strain with mutations in locus CP4 appeared (Table 3).

### 3.4. Biofilm Quantification

Based on dry weight measurements, all isolates were able to form biofilm. The biofilm production from each strain (P1, P2, P3, E1, and E2) is demonstrated in Figure 3. No statistical difference for biofilm quantification was found between these strains.

### 3.5. Electron Microscopy

The biofilms of *C. parapsilosis* strains P1 and E2 were observed using SEM (Figure 4). We confirmed the biofilms had produced an extracellular matrix (ECM). A comparison of the P1 biofilm with that of E2 showed no differences. The *C. parapsilosis* produced an extensive ECM with many fungal cells forming hump-shaped clumps. Additionally, there were many adhesive substances on the surface of the preparation. Our observation revealed the pseudohyphae were present only in limited areas, whereas thick and massive biofilms were formed in many areas.

### 3.6. Intervention by the ICT

When three nosocomial infections were detected in the NICU in one month, the ICT classified the occurrence as a *C. parapsilosis* outbreak. An environmental survey revealed that the isolated *C. parapsilosis* strains were the same. Additionally, the nurses caring for the patients (P1, P2, P3, and E2) were on the same team. We assumed that the *C. parapsilosis* infection had horizontally propagated through the hands of healthcare workers via the humidifier. The ICT had consulted the manual for cleaning incubators, but the incubator’s management method stipulated no procedure for patients who no longer needed humidification. The ICT suggested that NICU staff should check all the unused humidifiers of the incubators during daily rounds.

## 4. Discussion

We analyzed the biofilm of *C. parapsilosis* that caused the outbreak in the NICU. *C. parapsilosis* is one of the most common causes of fungal bloodstream infections, particularly for immunocompromised patients, and for patients who require the use of medical devices such as a central venous catheter. Biofilms are closely related to pathogenicity of *C. parapsilosis*. Additionally, the biofilms that *C. parapsilosis* form tend to spread to other medical devices, facilities, and liquids, and to the hands of healthcare providers [12,13,14,15]. In neonates, *C. parapsilosis* is the second most common yeast associated with bloodstream infections [3,4,5,6]. Additionally, several studies have reported that *C. parapsilosis* is a common source of outbreaks in NICUs [16,17,18,19]. All three patients with invasive fungal infections had underlying medical conditions and indwelling medical devices. The biofilm production of *C. parapsilosis*, including the strains isolated from the three patients with invasive infections, was evaluated by dry weight, which has been reported to correlate with biofilm adherence to medical devices [20]. Dry weight measurements showed that all *C. parapsilosis* isolates in this study produced biofilms, and there were no significant differences in quantification. *C. parapsilosis* in this study were proved to be the same strain by microsatellite analysis, so quantification of their biofilms showed no difference.

Morphological evaluation of biofilms is necessary, and confocal microscopy and electron microscopy are often used. In this study, we used SEM to analyze biofilm morphology using strains of isolates that had caused invasive fungal infections, and revealed a different biofilm morphology than previous studies. SEM observation showed many clumps of fungal cells with ECM. However, we observe few pseudohyphae. Regarding the biofilms of this study compared with those of past reports, we should note that the biofilms in this study formed very large masses with dense ECM, and they seemed quite sticky [12,21,22,23]. We theorized that *C. parapsilosis* in invasive fungal infections likely forms unusual biofilms with very dense ECM. These characteristics probably should be re-evaluated as highly pathogenic. According to Gomez-Molero et al., as surface adherence represents the first step in biofilm formation, non-smooth colonies are generally more adherent to medical devices than smooth colonies [6]. Our *C. parapsilosis* also showed non-smooth formation, so we thought our *C. parapsilosis* found it easy to stick to various devices and would easily horizontally transmit.

Another task is identifying the transmission route of an outbreak for preventing a next outbreak. *C. parapsilosis* is known to exist in hospital water environments, such as in sinks and in milk preparation rooms, which could also be sources of infection during the course of hospital care [24,25,26]. Additionally, *C. parapsilosis* cases have been described previously where the hands of HCWs turned out to be the predominant mode of transmission [27,28,29,30]. Therefore, we investigated mainly the water environment and hands of HCWs. As a result, we revealed the suspected route, which was verified when microsatellite analysis showed that the *C. parapsilosis* isolated from patients and environments was the same fungi. Despite the humidifier with *C. parapsilosis* having not been in use, it was still standing water in the absence of regular cleaning. Though preterm infants usually require humidification for a while after birth, it is unnecessary as skin matures. We assumed that the humidifier had not been cleaned after use. When the temperature in a humidifier reaches 100 °C during use, microorganisms, including fungi, tend not to propagate. Based on these results, we determined that the unused humidifier had caused the outbreak. However, as far as we could ascertain, there has been no report of an outbreak of *C. parapsilosis* originating in a humidifier. 

In recent years, clinicians have reported increases in instances of antifungal resistance to treatment with FLCZ in *C. albicans* and non-albicans fungi, and the MICs for echi-nocandin in *C. parapsilosis* have tended to be higher than those of other *Candida* species [31,32,33]. However, our three *C. parapsilosis* patients’ strains were susceptible to FLCZ and echinocandin. While we proved that our *C. parapsilosis* produced biofilm, and the biofilm formation had a characteristic structure, we did not measure ECM content and β-D glucan content in *C. parapsilosis* cell walls in biofilms. Cell walls, and specifically glucan change, associate with drug resistance, and several studies reported that the matrix β-D glucan of *Candida* species biofilms is one of the causes of drug resistance [34,35,36,37]. *C. grabrata* cells in biofilms seem to contribute more to the replacement of β-D glucan [36]. Additionally, the detection of the MNN2 gene in *C. grabrata* indicated biofilm matrix and cell wall variabilities that increased the resistance to antifungal drugs [37]. Though for *C. parapsilosis*, β-D glucan in its biofilm is associated with drug resistance, the resistance mechanism is unknown. As our three patients had high β-D glucan in blood cultures, and the biofilms of our *C. parapsilosis* produced ECM, analyzing the β-D glucan of our *C. parapsilosis* biofilm may have revealed the mechanism of drug resistance. We have not been able to investigate that yet, and we will leave it for future work. 

In order to prevent the spread of *C. parapsilosis,* infection control measures and laboratory testing systems similar to those used during outbreaks are necessary.

In conclusion, biofilms produced by *C. parapsilosis* cause invasive infections and nosocomial infections. If a horizontal infection of *C. parapsilosis* suspected, accurately tracing the route of infection is of paramount importance.

## Figures and Tables

**Figure 1 jof-08-00700-f001:**
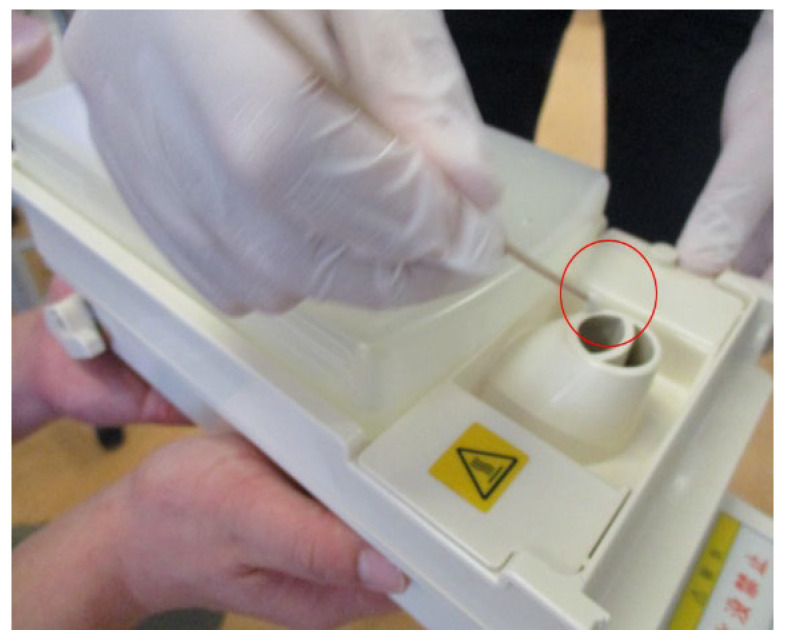
Location where the infection control team members took the environmental culture. The red circle is the tip of a swab. This picture shows the spot where the environmental culture was taken from the humidifier of the incubator (E2). The patient in E2 was fungemia negative.

**Figure 2 jof-08-00700-f002:**
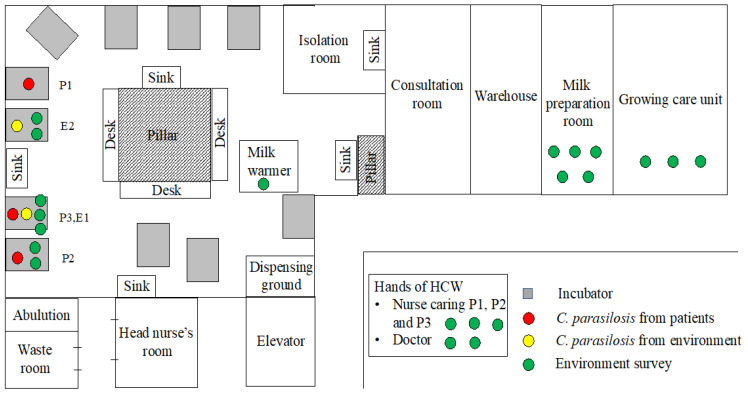
Schematic map of the ward layout of the Kurume University Hospital NICU and the locations where *C. parapsilsilosis* strains were isolated. The gray rectangles represent incubators. The red circles represent *C. parapsilosis* isolates, sampled from patients. The yellow circles represent *C. parapsilosis* isolates, sampled from the environment. The green circles represent the environmental cultures from the NICU. NICU: neonatal intensive care unit, HCW: health care worker.

**Figure 3 jof-08-00700-f003:**
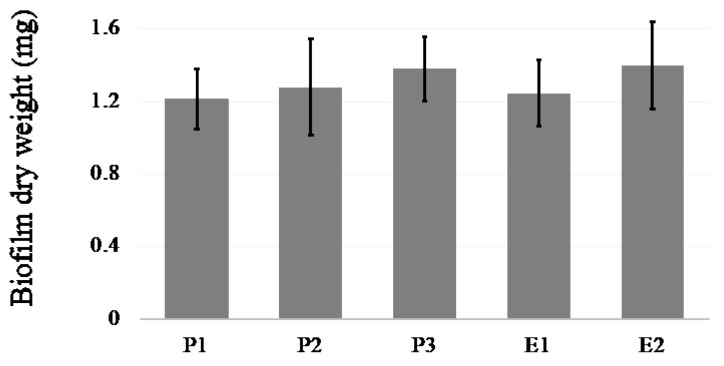
Biofilm quantification of *C. parapsilosis*.

**Figure 4 jof-08-00700-f004:**
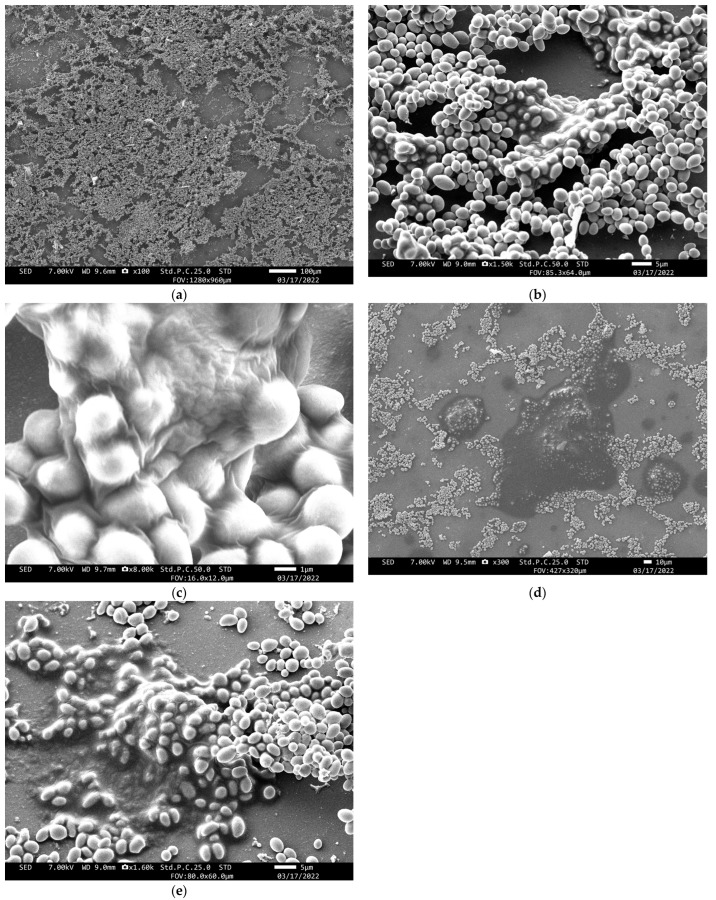
Biofilms of the *C. parapsilosis* strains were observed via scanning electron microscopy. The pictures (**a**–**c**) were taken of the P1 strain, and the pictures (**d**,**e**) were taken of the E2 strain: (**a**) the biofilm (100×); (**b**) many fungal cells formed hump-shaped clumps (1500×); (**c**) the surface of the clumps (8000×); (**d**) other points of the biofilm, and the liquid extracellular matrix produced on the surface of the preparation (300×); and (**e**) many fungal cells adhered to adhesive substances in the extracellular matrix (1600×).

**Table 1 jof-08-00700-t001:** The clinical characteristics of the 3 patients. PICC: peripherally inserted central catheter; ELBWI: extremely low birth weight infant; RDS: respiratory distress syndrome; NEC: necrotizing enterocolitis; L-AMB: liposomal amphotericin B; F-FLCZ: fosfluconazole; and MCFG: micafungin.

Patient No.	Gender	Gestational Age (Weeks)	Bith Weight (g)	Age of Onset (Days)	Basal Disease	Site of Infection	Date of Culture	Medical Device	Antifungal Therapy	β-D Glucan (pg/mL)
P1	female	34.5	2760	52	Preterm-infant Neonatal asphyxia Meconium peritonitis	blood	9 September 2020	PICC	L-AMB	507.4
P2	female	39.0	2872	44	VACTERL association	urine	26 September 2020	Urethral catheter	F-FLCZ	156.0
P3	male	24.5	783	47	ELBWI RDS NEC	blood	27 September 2020	PICC	MCFG	60.0

**Table 2 jof-08-00700-t002:** Antifungal agent susceptibility profiles (minimum inhibitory concentrations, mg/L).

Patient No.	AMPH-B	FLCZ	MCFG	VRCZ	CPFG
P1	1	0.5	0.5	≤0.01	1
P2	0.5	2	1	0.03	1
P3	1	0.5	1	≤0.01	1

AMPH-B: amphotericin B; FLCZ: fluconazole; MCFG: micafungin; VRCZ: voriconazole; and CPFG: caspofungin.

**Table 3 jof-08-00700-t003:** Microsatellite fragment analysis.

	Multi Locus Genotype (bp)			
Strain No.	CP1	CP4	CP6	B5
P1	244/244	305/324	283/312	108/110
P2	244/244	324 *	283/312	108/110
P3	244/244	305/324	283/312	108/110
E1	244/244	305/324	283/312	108/110
E2	244/244	305/324	283/312	108/110

* An allele for 324 bp in locus CP4 might be a null allele.

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
