# Peer review of "Characteristics of Biofilms Formed by C. parapsilosis Causing an Outbreak in a Neonatal Intensive Care Unit"

_jof, 2022, doi:10.3390/jof8070700_

Round 1

Reviewer 1 Report

This manuscript reports an outbreak of C. parapsilosis at a NICU and try to demonstrate the route of spread. The paper is simple and should be improved in some aspects, as described bellow:

In the Abstract, the sentence “Micro satellite analysis, dry weight measurements, and 23 observation by electron microscopy were performed against the strains.” is loose; the authors should give more context to it.

The topic 2.4 (Materials and Methods) should be renamed by “Identification test and Antifungal Susceptibility Profile”. Also, the authors should include the incubation temperature and the breakpoints used to evaluate the results of susceptibility test.

In the topic 2.5 (Materials and Methods), the authors should clarify the use of Microsatellite analyses and why they chose primers mentioned primers.

Candida species do not have conidia. So, in the topic 2.6 (Materials and Methods) and throughout the manuscript, the authors should change the word “conidia” by “yeasts” or “fungal cells”. Also, since the study was conducted with only five samples of Candida, I would like to know why authors did not use other classical and simple parameters to evaluate biofilm formation, like biomass and cell viability. I think these parameters would enrich the work and allow more robust comparisons with data on the literature.

In the topic 3.1 (Results), take off the “s” of “patients 1” and “patients 2”.

Figure 2, change “streins” by “strains”.

In the topic 3.4 (Results), no standard deviation was included in the description of biofilm dry weight. It is mandatory to include, and I think the authors should make a graph of this result.

Topic 3.5 (Results), “comparison” and not “comparision”.

The Discussion is poor and should be improved. The first sentence of Discussion should be reviewed. They also should discuss the differences observed in biofilm dry weights of the isolates.

Reviewer 2 Report

This work studies the outbreak of Candida parapsilosis in a NICU. C. parapsilosis were detected in 3 inpatients and in 2 environmental cultures (incubator, humidifier), and C. parapsilosis cultures were from the same strain. The results indicate that the outbreak had been a horizontal transfer through the humidifier of the incubator and that the C. parapsilosis had produced biofilm, which promoted an invasive and infectious outbreak.

The work is interesting and relevant.

Several points need to be adjusted and correctly described as there is a lack of critical details.

In detail:

-        It is the reason that C. parapsilosis prone to produce more biofilms than other Candida species”. C. albicans is most prevalent and the one that produces more biofilms – What is the meaning of this sentence? This is not very clear. Needs to be re-written/re-adjusted;

-        Ethical approval: date is missing. Please indicate it;

-        Some materials and reagents (e.g. sabouraud, antifungals) have no brand and country. Please check this in the entire MS and add this information;

-      Fihure 2 should have more quality;

-        Other Candida species are also high producers of beta-D glucan in their biofilms. This works are essential in this area and should be discussed by the authors in the Discussion section: 10.3390/biom8040130; 10.3390/genes9040205; 10.1128/AAC.02378-12; 10.1128/AAC.01056-06.

Reviewer 3 Report

This manuscript describes 3 cases of invasive C. parapsilosis infection among infants in a NICU.  Appropriately, an infection control investigation was activated in response to this mini-outbreak and identified a likely source.  The individual isolates appeared to be clonally related and generate biofilms in vitro.  Although the investigation of this outbreak is thorough and complete, there is very little to this manuscript that is novel or unexpected.  C. parapsilosis is well-documented to be present in hospital environments including NICUs, and horizontal transmission is likewise common and well documented.  The fact that the source involved humidification units with standing water that did not undergo routine maintenance is neither surprising nor novel.  Additionally, the manuscript suffers from atypical grammar and organizational flaws that make it difficult to read.  Additional comments for consideration as follows:

1. Line 41, "Infections in a...NICU are fundamentally hospital-acquired infections...." This statement really applies only to late-onset infections.  Early-onset infections are generally acquired from delivery and would not be amenable to environmental infection control.  This is an important distinction.

2. Line 45, "...C. parapsilosis prone to produce more biofilms than other Candida species."  This is an oversimplification and is misleading.  C. albicans, for example, is well known to produce extensive biofilms.  Additionally different isolates of the same species can also differ markedly in their biofilm production.

3. Section 2.3 - How were the sites that were part of the environmental survey identified?  Samples from the hands of the healthcare workers may have also been informative, particularly given the conclusion that this was the likely route of dissemination (which is also well documented in the literature).

4. Summary statistics provided in section 3.1 (% male, average age of onset, average weight) are not needed when n=3.

5. The Discussion is a series of topics that lack organization or theme. This section needs to be substantially revised and limited to putting the results in the context of current literature.

Round 2

Reviewer 2 Report

All my questions were properly addressed. Thank you. 

Reviewer 3 Report

The manuscript has improved based on the response to previous comments, although the English language and style remain unusual in numerous places. Concerns about novelty of the findings remain.